# First tomographic observations of gravity waves by the infrared limb imager GLORIA

Isabell Krisch[1], Peter Preusse[1], Jörn Ungermann[1], Andreas Dörnbrack[2], Stephen D. Eckermann[3], Manfred Ern[1], Felix Friedl-Vallon[4], Martin Kaufmann[1], Hermann Oelhaf[4], Markus Rapp[2,5], Cornelia Strube[1], and Martin Riese[1,6]

[1]Forschungszentrum Jülich, Institute of Energy- and Climate Research, Stratosphere (IEK-7), Jülich, Germany
[2]Deutsches Zentrum für Luft- und Raumfahrt, Institut für Physik der Atmosphäre, Oberpfaffenhofen, Germany
[3]E. O. Hulburt Center for Space Research, Naval Research Laboratory, Washington, D.C., USA.
[4]Karlsruhe Institute of Technology, Institute of Meteorology and Climate Research, Karlsruhe, Germany
[5]Meteorologisches Institut München, Ludwig-Maximilians-Universität München, Munich, Germany
[6]Institut für Atmosphären- und Umweltforschung, Bergische Universität Wuppertal, Wuppertal, Germany

*Correspondence to:* I. Krisch (i.krisch@fz-juelich.de)

**Abstract.** Atmospheric gravity waves are a major cause of uncertainty in atmosphere general circulation models. This uncertainty affects regional climate projections and seasonal weather predictions. Improving the representation of gravity waves in general circulation models, is therefore of primary interest. In this regard, measurements providing an accurate 3D characterization of gravity waves are needed. Using the Gimballed Limb Observer for Radiance Imaging of the Atmosphere (GLORIA), the first airborne implementation of a novel infrared limb imaging technique, a gravity wave event over Iceland was observed. An air volume disturbed by this gravity wave, was investigated from different angles by encircling the volume with a closed flight pattern. Using a tomographic retrieval approach, the measurements of this air mass under different angles allowed for a 3D reconstruction of the temperature and trace gas structure. The temperature measurements were used to derive gravity wave amplitudes, 3D wave vectors, and direction-resolved momentum fluxes. These parameters facilitated the backtracing of the waves to their sources on the south coast of Iceland. Two wave packets are distinguished, one stemming from the main mountain ridge in the South of Iceland, a second one from the smaller mountains in the North. The total, area-integrated fluxes of these two wave packets are determined. Forward ray-tracing reveals that the waves propagate laterally more than 2000 km away from their source region. A comparison of a 3D ray-tracing version to solely column based propagation showed that lateral propagation can help the waves to avoid critical layers and propagate to higher altitudes. Thus, the implementation of oblique gravity wave propagation into general circulation models may improve their predictive skills.

## 1 Introduction

Gravity waves (GWs) are oscillations in wind velocity and temperature with buoyancy as restoring force (Fritts and Alexander, 2003). They are the main driver of prominent circulation patterns in the mesosphere, such as the wind reversal in the mesosphere – lower thermosphere region (MLT). The exerted drag induces the meridional circulation in the MLT and finally leads to the cold summer mesopause and the warm winter stratopause (Holton, 1982, 1983; McLandress, 1998; Siskind, 2014).

GWs not only have a strong influence on the mesosphere but also on the stratosphere. There, they influence the circumpolar jet and its varying degrees of strength (McLandress et al., 2012; Ern et al., 2016); the tropical quasi-biennial-oscillation (QBO) of stratospheric tropical winds and its teleconnections into the extratropical troposphere (Dunkerton, 1997; Kawatani et al., 2010; Ern et al., 2014); and the variability in the strength of the meridional Brewer Dobson circulation (Alexander and Rosenlof, 2003; Butchart, 2014). These stratospheric circulations then have an impact on near-surface seasonal weather and regional climate via dynamical couplings with the troposphere (Scaife et al., 2016; Kidston et al., 2015).

Considering their small scales, the full spectrum of GWs cannot be resolved in general circulation models (GCMs) due to computational issues. Hence, they are simplified in form of parameterizations. Consequences of these GW parameterizations are for example surface temperature uncertainties of up to $2\,\mathrm{K}$ (Sigmond and Scinocca, 2010) and pressure discrepancies of several $\mathrm{hPa}$ at polar latitudes (Sandu et al., 2016) in climate projections. Improved weather predictions and climate projections therefore require more advanced parameterization schemes as proposed by various studies (Kim et al., 2013; Bushell et al., 2015; de la Camara and Lott, 2015; Amemiya and Sato, 2016).

One of the strongest simplifications, used for parameterizations, is to assume solely vertical propagation of GWs. However, several modelling studies, have highlighted the importance of 3D propagation of GWs to correctly reproduce the above mentioned circulation patterns (Sato et al., 2009; Preusse et al., 2009; McLandress et al., 2012; Kalisch et al., 2014; Ribstein and Achatz, 2016). Further, GW source distributions and launch parameters, such as propagation direction and wavelength spectrum, are often over-simplified in parameterizations (McFarlane, 1987; Hines, 1997; Alexander and Dunkerton, 1999; Scinocca and McFarlane, 2000; Beres et al., 2005; Richter et al., 2010; Garcia et al., 2017) and need validation by observations (Geller et al., 2013).

To underline the importance of 3D propagation and validate GW source distributions, measurements are needed, which allow for a full 3D wave characterization including the propagation direction (Alexander et al., 2010). Such a characterization is, in principle, possible from various in-situ techniques. Several methods were developed to evaluate data from close-to-vertical profiles taken by radiosondes, dropsondes or falling spheres. These methods include hodograph analysis (Guest et al., 2000), Stokes method (Eckermann and Vincent, 1989) or a combination of wind and temperature measurements in a common approach (Wang and Geller, 2003; Zhang et al., 2014). Furthermore, there are multiple techniques based on horizontal traces for example from airplane measurements (Alexander and Pfister, 1995; Fritts et al., 2016; Smith et al., 2016; Wagner et al., 2017) and observations by superpressure balloons (Boccara et al., 2008; Hertzog et al., 2008). All these methods have in common that they infer the wave direction via polarization and dispersion relations and do not reveal the 3D wave structure directly.

First 3D wave structures from satellite measurements in the stratosphere are presented by Ern et al. (2017) and Wright et al. (2017). However, these studies are based on nadir observations of the Atmospheric Infrared Sounder (AIRS) satellite instrument and are limited by the coarse vertical resolution. This implies that GWs with vertical wavelengths below $15\,\mathrm{km}$ are invisible to the instrument. In the mesosphere, a full wave characterization of short scale GWs has been achieved with the Middle Atmosphere Alomar Radar System (MAARSY; Stober et al., 2013). For medium scale GWs in the mesosphere, a full characterization has been derived by combining lidar and airglow imager measurements (Bossert et al., 2015; Lu et al.,

2015; Cao et al., 2016). However, all these observations are limited to a few ground based stations. Further, it is difficult to link observations at altitudes as high as the mesopause region to specific GW sources, which are usually located at much lower altitudes in the troposphere and lower stratosphere. So far no measurement technique existed to measure the 3D structure of mesoscale GWs in the lower stratosphere.

A novel technique to measure GWs in the upper troposphere – lower stratosphere, i.e. close to the GW sources, is limb imaging. Limb imaging allows for a 3D reconstruction of the atmospheric temperature and consequently a full characterization of mesoscale GWs. The development of the Gimballed Limb Observer for Radiance Imaging of the Atmosphere (GLORIA) is the first implementation of such an airborne infrared limb imager (Friedl-Vallon et al., 2014; Riese et al., 2014).

This technique was applied for the exploration of a GW for the first time in a research flight on 25 January 2016 above
Iceland. The results of this research flight are presented in this paper. Section 2.1 describes the instrument and the retrieval technique. The  results are presented in section 2.2 and subsequently analyzed for GWs in section 3.1. The obtained GW parameters are used for a wave propagation study in section 3.2.

## 2   Data and methodology

### 2.1   Measurement technique

This paper is based on tomographic measurements taken by the infrared limb imager GLORIA on board the German high altitude – long range research aircraft (HALO). The aircraft campaign took place from December 2015 to March 2016 with campaign bases in Kiruna, Sweden, and Oberpfaffenhofen, Germany. In total there were 21 research flights performed covering $20°N$ to $90°N$ and $80°W$ to $30°E$. The scientific targets of this campaign were to demonstrate the use of infrared limb imaging for gravity wave studies (GWEX), to study the full life cycle of a gravity wave (GW-LCYCLE), to investigate the Seasonality
of Air mass transport and origin in the Lowermost Stratosphere (SALSA), and to observe the Polar Stratosphere in a Changing Climate (POLSTRACC).

GLORIA combines a Michelson interferometer with a two-dimensional infrared detector and measures molecular thermal emissions in the spectral range between $780\,\mathrm{cm}^{-1}$ and $1400\,\mathrm{cm}^{-1}$ (7.1 to $12.8\,\mu\mathrm{m}$). It has a 256x256 pixels detector. However, to increase the read-out time, only a subset of 48x128 pixels is used. Thus, 6144 spectra are recorded simultaneously.
GLORIA's line-of-sight aims towards the horizon on the right side of the aircraft and measures infrared radiation emitted by molecules in the atmosphere. The point of the line-of-sight which is closest to the earth surface is called tangent point. Due to the curvature of the earth surface and the atmospheric density profile, the weighting function of the measurement signal has its maximum around this tangent point (Riese et al., 1999). This means that typically most of the measured radiation is emitted around the tangent points, which are located between $5\,\mathrm{km}$ and aircraft flight altitude. The horizontal observation angle of
GLORIA can be adjusted from $45°$ (right-forward) to $135°$ (right-backward) in respect to the aircraft's flight direction. In this way, the instrument can investigate the same air volume from different directions, which allows for a tomographic retrieval scheme (Ungermann et al., 2010; Kaufmann et al., 2015).

The basis of a tomographic retrieval scheme is a fast radiative transfer model. For the retrievals presented in this paper the Juelich Rapid Spectral Simulation Code Version 2 (JURASSIC2; Ungermann et al., 2010) is used. With this radiative transfer model $\boldsymbol{F}(\boldsymbol{x})$ infrared radiances can be calculated directly from an atmospheric state $\boldsymbol{x} \in \mathbb{R}^n$. Reconstructing the atmospheric state $\boldsymbol{x}$ from the infrared measurements $\boldsymbol{y} \in \mathbb{R}^m$ (the so called retrieval or inverse modelling) in contrast presents a non-linear

inverse problem, which is solved with an iterative minimization approach (Ungermann et al., 2011, 2015). For this, the cost-function

$$J(\boldsymbol{x}) = (\boldsymbol{F}(\boldsymbol{x}) - \boldsymbol{y})^T \mathbf{S}_\epsilon^{-1} (\boldsymbol{F}(\boldsymbol{x}) - \boldsymbol{y}) \tag{1}$$

has to be minimized. Here $\mathbf{S}_\epsilon \in \mathbb{R}^{m \times m}$ represents the covariance matrix of the measurement error $\epsilon$. To get a unique and well-constrained solution to this minimization problem, a regularization term is added to the cost-function (Ungermann et al.,

2010). This term ensures that the solution is physically reasonable.

As a-priori field $\boldsymbol{x}_a$ a temperature field from the European Centre for Medium-Range Weather Forecasts (ECMWF) operational analyses at resolution T1279/L137 was used, which was smoothed in all spatial directions to remove GW signatures. This smoothing was done by applying a low-pass Fourier filter with cut-off wavenumber 18 in zonal direction. In height and latitude direction a Savitzky-Golay (SG) filter (Savitzky and Golay, 1964) was applied with 4th order polynomials over 11 and

15 25 neighbouring points respectively. On the one hand, the so generated a-priori field improves the convergence speed of the iterative minimization, as this temperature structure is close to the true values due to the high quality of the ECMWF model. On the other hand, the smoothening ensures that any GW signature in the retrieval result does not stem from the used a priori data. If the a-priori data exerts any influence, it would dampen the GW structure.

For the present retrieval we used the spectral ranges listed in Tab. 1. In these spectral ranges the main emitters are $CO_2$,

$CCl_4$, $HNO_3$, and $O_3$. The volume mixing ratio of $CO_2$ is well known in this altitude range. Therefore, spectral lines of $CO_2$ are used effectively for the retrieval of temperature. Tomographic reconstructions of the 3D temperature distribution and the mixing ratios of $CCl_4$, $HNO_3$, and $O_3$ in the upper troposphere and lower stratosphere were achieved. However, a discussion of the trace gas distributions exceeds the scope of this paper.

An error analysis of the retrieval has been performed following the methods described in Ungermann et al. (2015). The

25 precision (noise error) is below $0.05\,\mathrm{K}$ and the accuracy, which includes misrepresented background gases, uncertainties in spectral line characterisation, uncertainties in instrument attitude, and calibration errors, is in the order of $0.5\,\mathrm{K}$.

The horizontal and vertical resolutions can be defined by the axes of the smallest ellipsoid containing all elements of the averaging kernel larger than half the maximum. Accordingly, in the middle of the performed hexagonal flight path, the vertical resolution is around $200\,\mathrm{m}$, the horizontal resolution around $20\,\mathrm{km}$.

The time needed to accomplish the hexagon was about $2\,\mathrm{h}$. During this time 2200 infrared images and corresponding spectra were taken. The presented tomographic retrieval represents a temporal mean over all these measurements.

**Table 1.** Spectral windows used for the retrieval presented in this paper. The last column indicates the retrieved quantity for each spectral range.

| | spectral range / $cm^{-1}$ | | | used for |
|---|---|---|---|---|
| 1 | 790.625 | – | 792.500 | temperature |
| 2 | 793.125 | – | 795.000 | $CCl_4$ |
| 3 | 796.875 | – | 799.375 | $CCl_4$ |
| 4 | 883.750 | – | 888.125 | $HNO_3$ |
| 5 | 892.500 | – | 896.250 | $HNO_3$ |
| 6 | 900.000 | – | 903.125 | $HNO_3$ |
| 7 | 918.750 | – | 923.125 | $HNO_3$ |
| 8 | 956.875 | – | 962.500 | temperature |
| 9 | 980.000 | – | 984.375 | temperature, $O_3$ |
| 10 | 992.500 | – | 997.500 | temperature, $O_3$ |
| 11 | 1000.625 | – | 1006.250 | temperature, $O_3$ |
| 12 | 1010.000 | – | 1014.375 | temperature, $O_3$ |

## 2.2 Research flight above Iceland

On the measurement day 25 January 2016, a southerly wind made landfall on the south coast of Iceland (Fig. 1), thus exciting mountain waves. These waves were predicted by the ECMWF forecast to be stationary above Iceland for more than 6 hours. Above 10 km altitude the zonal wind increased drastically with height and turned from southerly to south-westerly direction. This created a strong vertical wind shear, which influenced the propagation of the excited mountain waves. The wave structure over eastern Iceland was encircled by a hexagonal flight pattern with 460 km diameter between 10 and 12 UTC (Fig. 2). The aircraft flight altitude during this time was between 12.5 km and 13.5 km. Towards low altitudes, the GLORIA measurements were limited by clouds reaching up as far as 9 to 10.5 km. Before the hexagon a linear flight through the wave field has been performed to collect in-situ data at flight altitude and to release dropsondes.

For the GLORIA retrieval only the measurements taken between 10 and 12 UTC have been taken into account. To identify GWs in the retrieved 3D temperature field, the large scale temperature background, which is caused by the balanced flow and the stratification of the atmosphere, has to be separated from the smaller scale temperature variations caused by the GWs. This was done through applying SG filters with 3rd order polynomials over 25, 60, and 60 neighbouring points in vertical, zonal, and meridional direction. The values of these polynomials at the respective points are treated as temperature background. The remaining temperature residuals clearly reveal the complex structure of the wave field, which is demonstrated in 3D in Fig. 2.

In Fig. 3, horizontal and vertical cross sections through the measurement volume are presented. They show how the wave structure varies with height and horizontal location. For instance, the wave fronts directly above Iceland (64°N to 65.5°N and 14°W to 18°W are aligned east-west and tilted southwards against the prevailing southerly wind (Fig. 1). Further to the

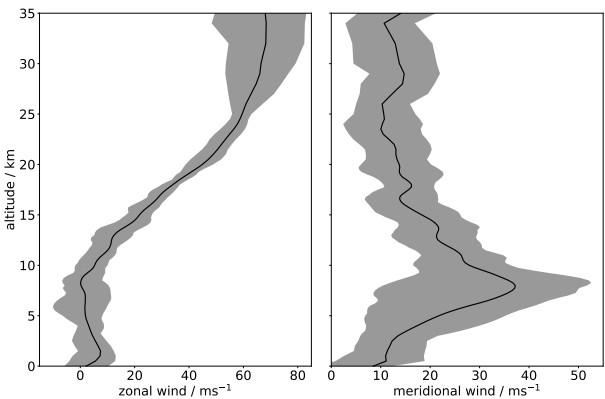

**Figure 1.** Mean zonal and meridional wind profiles from ECMWF operational analyses T1279/L137 above Iceland and the measurement region at 12 UTC on January 25, 2016. The grey area marks the spread of the wind profiles in this area.

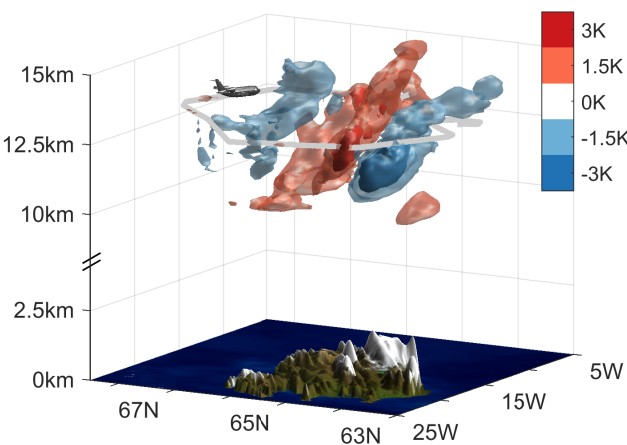

**Figure 2.** Tomographic retrieval of the temperature field for the research flight on January 25, 2016, over Iceland. Shown are isosurfaces of the temperature residual. The grey line around the retrieved 3D pattern indicates the flightpath.

north-east ($65°$N to $67°$N and $10°$W to $14°$W), the horizontal orientation of the wave fronts turns more into south-west to north-east. The horizontal wavelength varies inside the hexagon from $100\,\mathrm{km}$ up to $350\,\mathrm{km}$. The vertical wavelength of the waves is between $3\,\mathrm{km}$ and $6\,\mathrm{km}$. The temperature residuals range from $\pm4\,\mathrm{K}$ (in the south-west of the hexagon at an altitude of $12\,\mathrm{km}$, $64°$N to $65.5°$N, and $14°$W to $18°$W) down to $\pm1\,\mathrm{K}$ (in the smaller scale waves in the north-western part of the hexagon at $66°$N to $68°$N and $16°$W to $20°$W).

Fig. 4 shows a comparison of the retrieval results with in-situ measurements and ECMWF operational analyses with T1279/L137 resolution. The retrieval results and model data were interpolated onto the in-situ measurement locations. The GLORIA measurements agree well with the in-situ measurements. Some very short scales are beyond the spatial resolution of

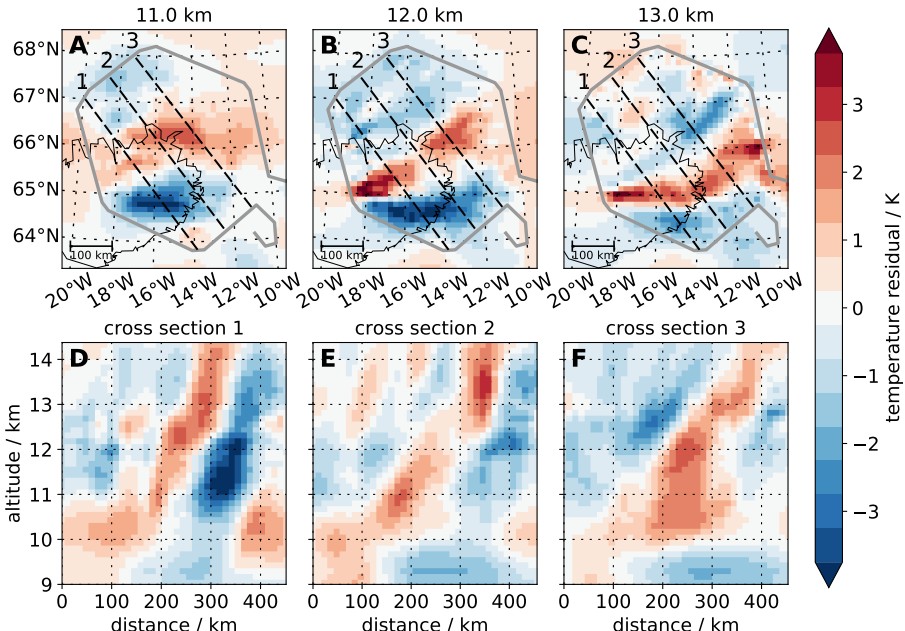

**Figure 3.** Horizontal (A-C) and vertical (D-F) cross sections through the 3D volume shown in Fig. 2. The grey line marks the flightpath. The locations of the vertical cross sections are indicated by numbered dashed lines.

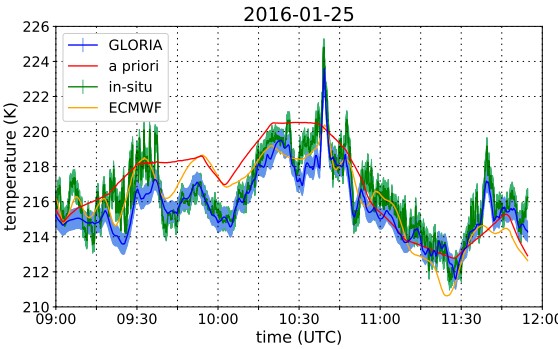

**Figure 4.** A comparison of the GLORIA retrieval results to in-situ temperature measurements and ECMWF operational analyses T1279/L137 at 12 UTC. The GLORIA retrieval and the ECMWF data were interpolated in space onto the flight path.

GLORIA. The ECMWF analysis catches the main variations, but the temperature oscillations are not as strong as in reality. GLORIA can better reproduce peaks as for example the one around 10:40 UTC. This comparison underlines the high quality of the GLORIA measurement data.

# 3 Results

## 3.1 Sinusoidal wave fits

In order to further interpret the GW structure and fully characterize it, wave parameters are derived using a small-volume few-wave decomposition technique (Lehmann et al., 2012). The algorithm performs 3D sinusoidal fits in small data cubes. In such a way, the wave amplitude, horizontal and vertical wavelengths, and 3D wave direction can be derived. In contrast to a Fourier transform, this technique allows for the characterization of waves with wavelengths larger than the cube size (up to a factor of 2.5 times the cube size). Due to the prevailing wavelength range in our measurements (cf. Fig. 2), a cube size of 160 km x 160 km x 3.6 km, containing 4900 data points, was chosen. The fitted parameters are the 3D wave vector $\boldsymbol{k} = (k, l, m)$ and the amplitudes (Fig. 5C). Horizontal and vertical wavelengths, and the horizontal wave direction were calculated from the wave vector $\boldsymbol{k}$ and shown in Fig. 5A, 5B, and 5E, respectively. Fig. 5F shows the orography of Iceland. The main mountain ridge is oriented in east-west direction. As expected for a mountain wave, the horizontal wave direction (Fig. 5E) is perpendicular to the ridge orientation. A discussion of relevant effects for the uncertainty of the fitted parameters and the resulting confidence intervals is given in Appendix A.

A key quantity of GWs is the vector of vertical flux of horizontal pseudo-momentum (short GW momentum flux, GWMF)

$$\boldsymbol{F_{ph}} = \frac{1}{2}\rho \frac{\boldsymbol{k_h}}{m}\left(\frac{g}{N}\right)^2\left(\frac{\hat{T}}{T}\right)^2 \tag{2}$$

where $\rho$ represents the air density, $\boldsymbol{k_h} = (k, l)$ the horizontal and $m$ the vertical component of the wave vector, $g$ the standard gravity, $N$ the buoyancy, $T$ the background temperature, and $\hat{T}$ the temperature amplitude (Ern et al., 2004). Low and high frequency terms are omitted here due to simplicity. Deviations from the full equations derived by Ern et al. (2004) are less than 1% in the observational range of GLORIA. For a full discussion of the relevance of all correction terms see the supporting information in Ern et al. (2017).

Integrating the GWMF over the horizontal extent of a GW event leads to the total momentum, which determines the maximal drag this GW event can exert on the background flow in coupling and dissipation processes. The fitted wave parameters in Fig. 5A - C are used to calculate the GWMF (Fig. 5D). The horizontal distribution of the GWMF clearly highlights two distinct wave packets: one with local GWMF of up to 50 mPa north of 66.2 °N and one with local GWMF of up to 100 mPa south of 66.2 °N. The GLORIA observations provide the horizontal variations of GWMF at 11.5 km altitude. This allows to integrate over the corresponding area of the two events and calculate the total momentum, a measure for the maximal drag this GW event can exert on the background flow in coupling and dissipation processes. This is a main advantage with respect to 1D wind observations, which can provide peak GWMF values but not the area for which these values are valid. The wave packet further south has a total momentum of 2.7 GN, the second wave packet further north only 0.4 GN. The total momentum of all the measured GWs above Iceland is 3.1 GN.

To classify this event, a comparison of all GW events in January 2016 has been performed in the 6-hourly operational analyses of ECMWF. First the temperature background was isolated, as described in Sec. 2.1 for the a-priori field, and subtracted from the original field. The remaining temperature residuals were analyzed for GWs using the 3D sinusoidal fit algorithm de-

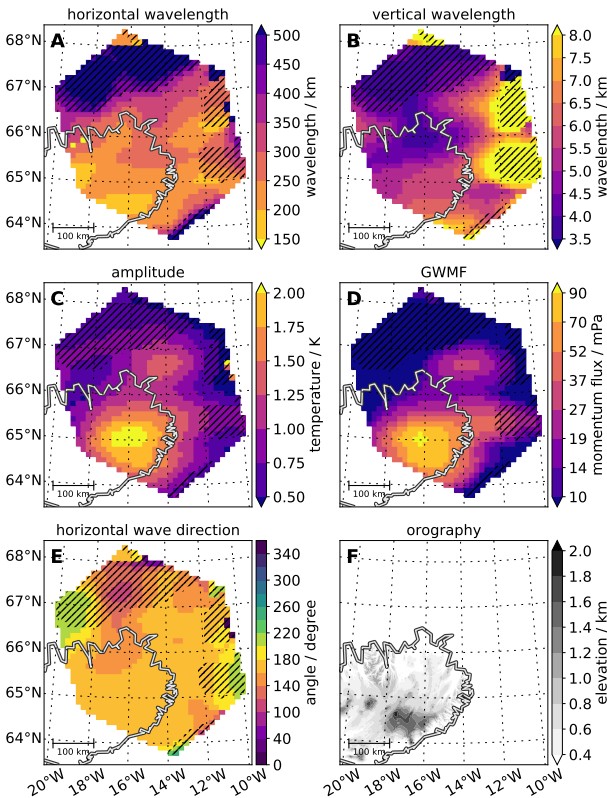

**Figure 5.** 3D sinusoidal wave fit of the GLORIA measurements in fitting cubes of $160\,\mathrm{km}$ x $160\,\mathrm{km}$ x $3.6\,\mathrm{km}$ at a center height of $11.5\,\mathrm{km}$. Non-significant fitting results with wavelengths above 2.5 times the cube size are hashed. These parameters are used to drive the GROGRAT model, the results of which are shown in Fig. 7. Panel E shows the direction of the horizontal wave vector. Eastward direction corresponds to $90\,^{\circ}$, southward direction to $180\,^{\circ}$.

scribed above. The GWMFs for all cubes were calculated. The GWMFs from all 124 analyses fields were combined to obtain the probability of GW occurrence (Fig. 6). Here, all GWMF values were considered independent of the horizontal and vertical wavelengths. Removing wavelengths larger than 2.5 times the cube size in order to filter less significant fits (not shown) induced no major changes in the general shape of the distribution. This indicates that GW events with less certain fits do not

5  bias the probability distribution.

For the GW event over Iceland similar GWMF magnitudes were determined from the ECMWF analyses and from the GLORIA measurements. Thus, a comparison of the measurement results with the occurrence probability determined from the ECMWF analyses seems reasonable. According to Fig. 6 the measured GW event can be classified as a very strong case since the sum of all occurrence probabilities of stronger events is well below $1\%$. This occurrence frequency is in good agreement

10  with Alexander et al. (2010), Hertzog et al. (2012), and Podglajen et al. (2016) who present satellite and super-pressure balloon measurements at slightly higher altitudes.

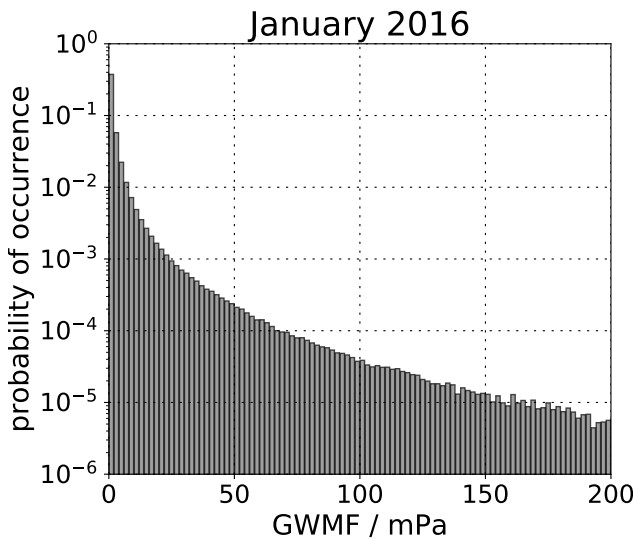

**Figure 6.** Probability of occurrence for GWs with specific momentum flux at $11.5\,\mathrm{km}$ altitude in a latitude band between $60°\mathrm{N}$ and $70°\mathrm{N}$ in January 2016 calculated from 6-hourly ECMWF operational analyses fields.

## 3.2 Wave propagation with GROGRAT

In order to identify the GW source, we used the Gravity wave Regional Or Global RAy Tracer (GROGRAT; Marks and Eckermann, 1995). GROGRAT describes the propagation of wave packets based on linear wave theory. Backward ray-tracing has been used in previous studies to locate GW sources (Preusse et al., 2014; Pramitha et al., 2015). In order to initialize a ray-tracer, the wave must be fully characterized. This capability is the main improvement of the GLORIA observations compared to previous remote sensing observations of temperature. GW parameters obtained from single vertical temperature profiles, lead to a cone of potential source regions instead of a precise source location (Gerrard et al., 2004). This is the reason why GWs derived from conventional limb scanner measurements have not been interpreted in terms of backward ray-tracing. Only the 3D nature and accuracy of the GLORIA measurements allows backtracing to the precise source location. This is further highlighted by the error analysis presented in appendix A.

In the error analysis, a systematic low bias of the vertical wavelengths was found, which is caused by the sinusoidal fit (appendix A). Therefore, the vertical wavelengths from the sinusoidal fits were scaled by a factor 1.1, according to the determined bias, before being used for the ray-tracing. For the propagation of GWs in a ray-tracing model temporally and spatially varying background temperature and wind fields are needed, which were obtained from ECMWF operational analyses.

Fig. 7 shows the backward ray-traces of the measured GWs from their measurement position (black crosses) down to the source location (red dots). The measurement position has been defined as the center point of the sinusoidal fitting cube. The strength of the GW is expressed by the size of the red dots, which has been chosen according to the GWMF at the source location. These GWMF values are conservative estimates, as the backward ray-tracing cannot account for dissipation

processes. The source locations of the GWs, and in particular those of the highest GWMF, gather around the main mountain ridge of Iceland. The GWs are, thus, likely to have been excited by the southerly wind approaching these mountains. The ray-traces from the wave packet measured further in the north partly stop in the north of the island at single mountain peaks.

As can be seen in Fig. 7C, the ray-traces need between 3 and 6 hours to reach the ground. This is in good agreement with a vertical group velocity of 2 to 3 km/h, which has been calculated from the measurements. Hence, the GWs are probably excited roughly 6 hours before the measurements were taken.

Forward ray tracing is used to examine the propagation of the GWs away from the measurement location (Fig. 7B). On the measurement day, the southerly wind turned into a strong westerly direction above 10 km, creating a strong vertical wind shear. In this wind shear the GWs started to propagate eastward. This is confirmed by the measurements: at 11 km (Fig. 3A) the GWs are mainly located above the eastern part of Iceland, while at 13 km (Fig. 3C) the wave fronts already stretch far across the ocean. The waves require about one day to propagate to an altitude of 20 km (Fig. 7C). At the same time, they travel horizontally more than 2000 km (Fig. 7B). Over Eastern Europe, the GWs are refracted due to a horizontal wind shear and, thus, change their horizontal wave vector from southward to westward. This allows the waves to quickly propagate upward into the westerly wind in the mid stratosphere.

To mimic a typical GW parameterization scheme used in GCMs (McLandress, 1998), a second GROGRAT run (1D-GROGRAT) was performed with solely vertical propagation, time-independent background, and a horizontal wave direction constant with respect to altitude. In contrast to the full GROGRAT version (Fig. 7C), where the GWs propagate into the mid stratosphere, the GWs in the simplified version dissipate below 20 km (Fig. 7D). Two processes might play a significant role here: First, in the 1D GROGRAT version the GWs are not refracted and the wave vectors do not change its horizontal orientation with altitude. The westerly background winds at higher altitudes do not favor the propagation of GWs with wave vectors perpendicular to the wind direction. Second, in the full GROGRAT run, the GWs propagate horizontally away from the source. Hence, the GWs avoid the critical level positioned above the source location and more GWMF is transported to higher altitudes. Global mountain wave modeling (Xu et al., 2017) suggests that this effect may prevail also on a global basis.

Neither a realistic orientation of the wave vector, nor oblique GW propagation are incorporated in GW parameterizations used in current climate and weather prediction models (McLandress, 1998; Alexander and Dunkerton, 1999; Richter et al., 2010; McLandress et al., 2012; Garcia et al., 2017). However, both processes are context of several studies aiming to improve GW parameterizations (Preusse et al., 2009; Sato et al., 2009; Kalisch et al., 2014; Amemiya and Sato, 2016; Ribstein and Achatz, 2016; Garcia et al., 2017). The present paper provides a strong motivation to finally implement these processes in current climate and weather prediction models. Especially, as this could close gaps of GWMF in regions with sparse sources (McLandress et al., 2012) and reduce the cold-pole bias of climate and weather prediction models in the lower stratosphere (Garcia et al., 2017).

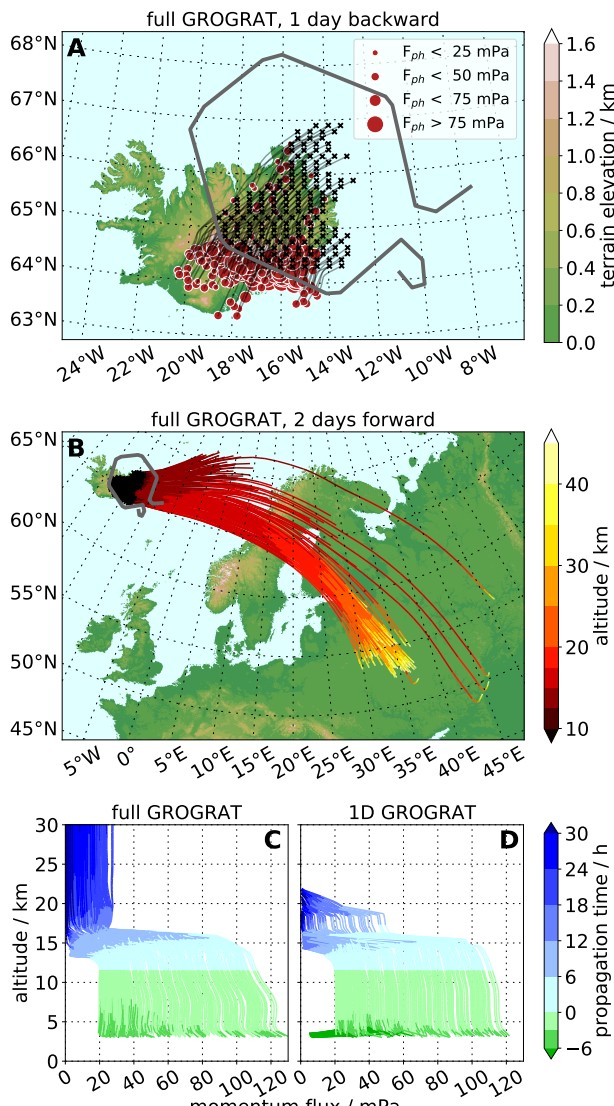

**Figure 7.** Ray traces calculated using the GROGRAT model. The starting positions of the rays are marked with black crosses and the grey line indicates the flight path. The size of the red circles in A indicates the GWMF at the end of the ray. Panel A shows the backward ray traces and panel B the forward ray traces, all starting at the measurement locations. Panels C and D show the change of GWMF with height for a full 4D GROGRAT (C) model run and a solely vertical 1D run (D).

## 4 Conclusions

In this paper, we presented the first tomographic measurements of temperature perturbations induced by a GW event. The 3D measurements recorded by GLORIA, the first airborne implementation of a novel limb imaging technique, enabled the

deduction of direction-resolved GWMF and the identification of two distinct wave packets. The retrieved 3D wave vectors were used as input in the ray-tracing model GROGRAT, which highlighted the orography of Iceland as the most likely GW source. Furthermore, upward from $11\,\mathrm{km}$ the wave packets propagate obliquely as is seen from the observation and reproduced by the ray-tracer. A comparison between the full GROGRAT model and a simplified 1D version indicated the relevance of oblique propagation for the GWMF deposition height. In the simplified version, all GWs deposited their momentum at an altitude of around $20\,\mathrm{km}$, whereas in the full version, waves were able to vertically propagate to the top of the model at $45\,\mathrm{km}$ and horizontally more than $2000\,\mathrm{km}$ away from their source, thus redistributing GWMF significantly. Given that weather prediction and climate models routinely use 1D models of GW propagation, the present findings demonstrate that considering 3D propagation could lead to significant improvements in weather forecasting and climate prediction.

*Data availability.* The tomographic retrieval data is available on the HALO database (https://halo-db.pa.op.dlr.de/).

*Competing interests.* The authors declare that they have no conflict of interest.

*Acknowledgements.* This work was partly supported by the Bundesministerium für Bildung und Forschung (BMBF) under project 01LG1206C (ROMIC/GW-LCYCLE), as well as by the European Space Agency (ESA) under contract 4000115111/15/NL/FF/ah (GWEX) and the Deutsche Forschungsgemeinschaft (DFG) project PR 919/4-1 (MS-GWaves/SV), which is part of the DFG researchers group FOR 1898

(MS-GWaves). The retrievals were performed on the JURECA supercomputer at the Jülich Supercomputing Center (JSC) as part of the JIEK72 project. The results are based on the efforts of all members of the GLORIA team, including the technology institutes ZEA-1 and ZEA-2 at Forschungszentrum Jülich and the Institute for Data Processing and Electronics at the Karlsruhe Institute of Technology. We would also like to thank the pilots and ground-support team at the Flight Experiments facility of the Deutsches Zentrum für Luft- und Raumfahrt (DLR-FX). Special thanks go to Andreas Diez at DLR-FX for providing the in-situ temperature data shown in Fig. 4 and to Corwin Wright

at the Centre for Space, Atmospheric and Oceanic Science at University of Bath for providing the plotting code for Fig. 2.

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

## Appendix A: Error analysis

In this section the effects of different errors onto the ray-tracing results presented in Section 3.2 are discussed. These errors may, in principle, be caused during each of the three main processing steps: temperature retrieval, background removal and sinusoidal wave fits (S3D). Retrieval errors can be divided into precision and accuracy (cf. Section 2.1). Due to the high number
of independent data in each S3D cube, the precision error (mainly due to noise) can be neglected, in particular since in this paper only GW events with amplitudes above a threshold of 0.5 K are considered. The error sources which lead to the accuracy error are systematic and slowly varying. Thus, their impact is mostly mitigated by the background removal.

The background removal separates the data into large scale variations and short scale fluctuations, the latter interpreted as GWs. The main effect of an unfavorably tuned background removal would be to eliminate real GWs. However, it would not
introduce errors in the fitted wave vectors. Thus, this has to be considered in a comparison with other data, but is not included in the further error discussion.

The third step, the S3D method, is based on the assumption that the fitting volume is filled by a homogeneous wave with a constant wave amplitude and a constant wave vector over the fitting volume. The discussion in sections 2.2 and 3.1 demonstrates that this assumption is only valid to a certain degree. In particular, we notice that the direction of the horizontal wave vector
and the vertical wavelength change with height as the wave is refracted by a changing background wind.

We use the results of the ray-tracer to estimate errors due to this change over height within the fitting volume. In Fig. A1 left column the instantaneous value $\xi_{z=11.5}$ of the ray at the middle point of each fitting volume is compared to an average $\bar{\xi}$ of all values of the ray in the height range of the respective S3D fitting volume (comparable to the S3D fitting result); here $\xi$ stands for either the vertical or horizontal wavelength or the horizontal wave direction. The mean vertical wavelength shows a systematic
low bias of around 10% compared to the instantaneous value in the middle (Fig. A1 A). This effect is taken into account and all vertical wavelengths from the sinusoidal fit (Section 3.1) are scaled with a factor of 1.1 before being used in the ray-tracing analysis (Section 3.2). For the horizontal wavelength (Fig. A1 C) and the horizontal wave direction (Fig. A1 E) no systematic bias could be identified.

As mentioned before, the accuracy with which the input wave parameters $\xi$ are determined is of high importance. This is
25 highlighted through varying the values $\xi$ by factor $\epsilon_\xi$ and comparing the ray-tracing results with a reference run. The variations $\epsilon_\xi$ for the vertical wavelength and the horizontal wave direction are chosen to be half the difference of the wave parameters at the upper ($\xi_{z=max}$) and lower ($\xi_{z=min}$) boundary of the S3D fitting volume as determined by the ray-tracing reference run (Fig. A1 B, F). The horizontal wavelength does not change much over the height of the fitting volume (Fig. A1 D). However Fig. 3 and Fig. 5 indicate a significant variation of the horizontal wavelength over the horizontal extent of the fitting volume.
Hence, for the error estimate ray-tracing calculations we chose an error value of $\pm15\%$ as estimate for the horizontal variation of the horizontal wavelength within the S3D fitting volume. In Tab. A1 the used error estimates for the three wave parameters are summarized.

The results of the back-tracing runs with wave parameters varied by the error estimates in Tab. A1 are shown in Fig. A2. Longer vertical wavelengths (Fig. A2 C-D) lead to more northward located sources, while rays from shorter vertical wave-

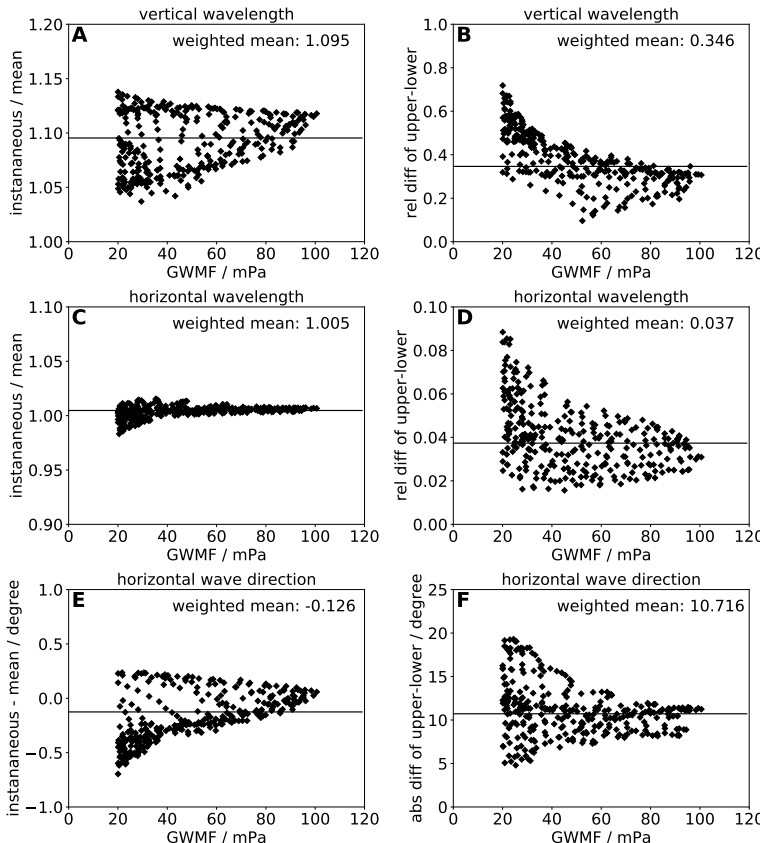

**Figure A1.** Comparison of mean values over the whole S3D fitting volume and instantaneous values in the middle for different wave parameters (left column) and variation of wave parameters from the lower to the upper boundary of the S3D fitting volume (right column). For all graphs GWMF weighted means are calculated and depicted as black lines.

**Table A1.** Error estimates for the wave parameters inferred by the S3D method based on the change of the parameters over the extent of the fitting cube.

| | |
|---|---|
| error estimate of the vertical wavelength | ±17% |
| error estimate of the horizontal wavelength | ±15% |
| error estimate of the horizontal wave direction | ±5.5° |

lengths (Fig. A2 B) end southward, i.e. upstream of Iceland over the ocean. This is due to the fact, that longer vertical wavelengths are associated with higher horizontal phase velocities and hence higher horizontal group velocities. Accordingly, in the case of shorter vertical wavelengths the waves are not able to compensate the background wind velocity and would origin from an upstream source. Actually, we find that the inferred bias (10% larger values for the vertical wavelengths), when corrected, improves the match of the ray-positions with the topography (Fig. A2 C).

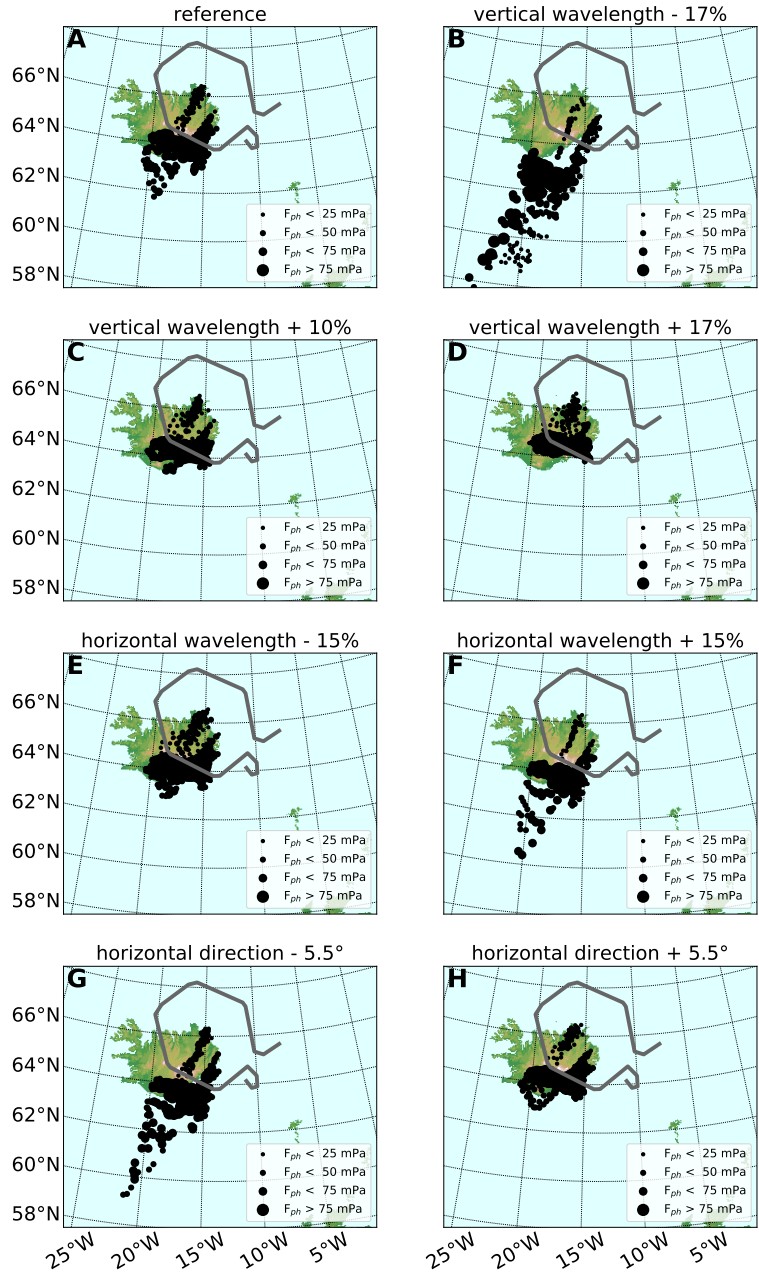

**Figure A2.** Backward ray-tracing with varying input wave parameters. The black dots mark the ray positions at 3 km altitude.

The horizontal wave direction has similar impact: When the wave is turned more into the south-easterly background flow (Fig. A2 H) the ray-paths are more vertically oriented and therefore reach the ground closer to the measurement volume. When they are turned away from the background wind (Figure A2 G), the intrinsic group velocity and the background wind are at an angle, the intrinsic group velocity does not fully compensate the background wind, and the waves cover a larger horizontal distance reaching onto the ocean upstream of Iceland.

Similar variations for the forward ray-traces are shown in Fig. A3. In all cases except for shorter vertical wavelengths, a major group of ray-traces reaches the model top at 45km altitude (white dots), i.e. our main findings presented in Sec. 3.2 are robust.

The error estimates demonstrate that a correct identification of the GW source is only feasible for highly accurate wave characterization, such as achieved here thanks to the high spatial resolution and accuracy of GLORIA 3D temperature measurements.

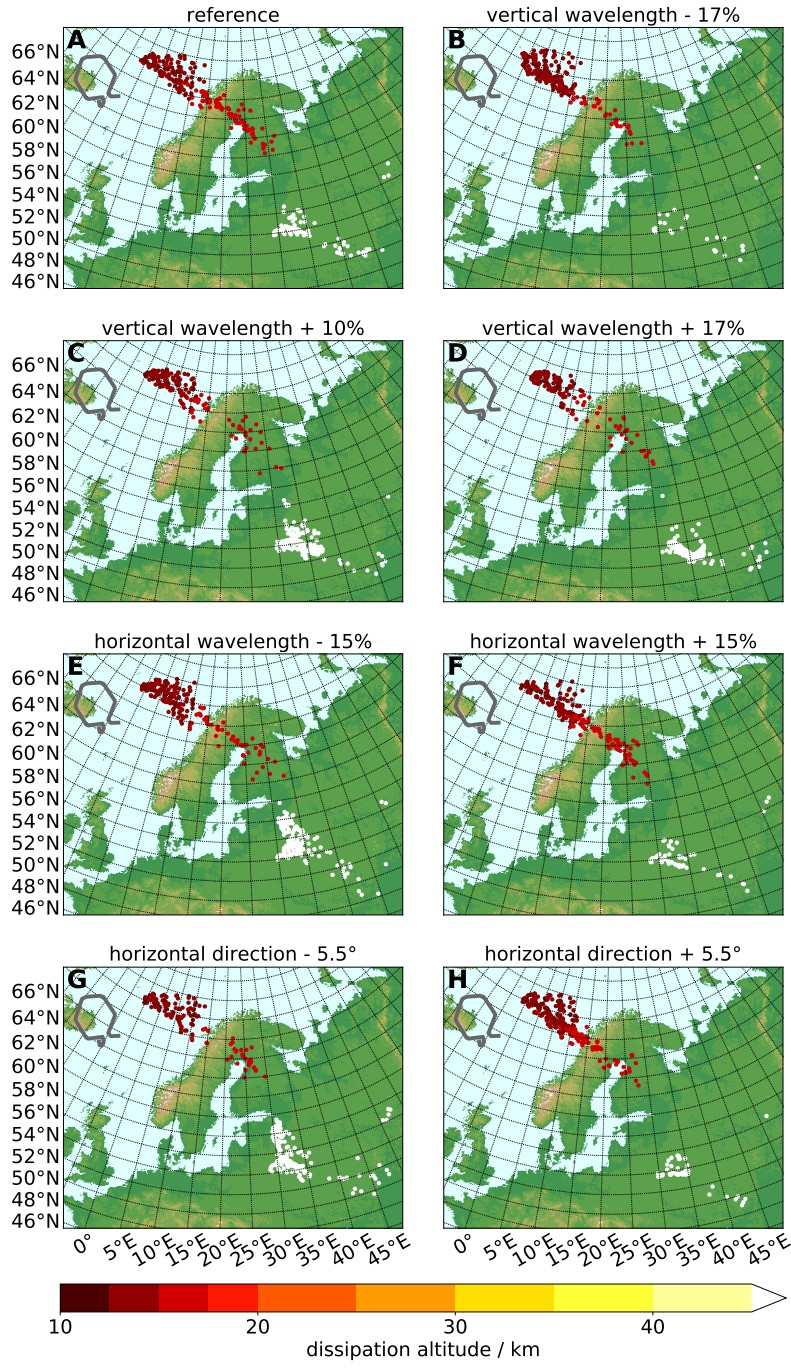

**Figure A3.** Forward ray-tracing with varying input wave parameters. The dots mark the dissipation point of the ray, the colour indicates the dissipation height. White dots mark waves which reach the model top at 45 km altitude.