# Peer review of "First tomographic observations of gravity waves by the infrared limb imager GLORIA"

_Atmospheric Chemistry and Physics, 2017_

## Referee Comment (RC1) · O. M. Christensen (Referee) · 31 Aug 2017

**1  General comments**

The manuscript describes the analysis of temperature data measured by the GLORIA instrument during a flight over Iceland conducted January 2016. The authors describe how they retrieve gravity wave (GW) parameters (wavelengths, amplitudes and momentum flux) from this data using ECMWF operational analysis for background wind estimation.

Based on this data the origins and future propagation paths of the waves are determined using a ray-tracing model. Thanks to the 3D nature of the tomographically

retrieved data this can be done with a much higher accuracy than previous studies, illustrating the importance of 3D reconstruction for studying GWs.

Finally the authors comment that the waves measured deposit their momentum at horizontal distances over 1000 km from their origin. They show that the altitude at which this momentum is deposited in their model depends on whether this horizontal transportation is taken into account. This illustrates the importance of this transport for correctly modelling the influence of GWs on the dynamics of the atmosphere.

In general I find that the manuscript is well written, is interesting and that the underlying data and analysis methods (including error analysis) support its conclusions, and thus will recommend its publication with some minor comments/questions from my side.

**2   Specific comments**

**P4L7:** *"As a-priori field $x_a$ a temperature field from the European Centre for Medium-Range Weather Forecasts (ECMWF) operational analyses at resolution T1279/L137 was used, which was smoothed in all spatial directions to remove GW signatures."*

What smoothing filter was used (type and fwhm)?

**P4L4:** *"The vertical resolution can be defined as the full width at half maximum of the averaging kernel matrix and is around 200 m at an altitude of 11.5 km"*

Are you simply looking at the elements in the row of the AVK matrix corresponding to the grid points directly above the grid-point of the row, or are you collapsing (i.e. performing a summation) the two other dimensions (x,y) before this FWHM is calculated?

If the first approach is used, some description of the elements located diagonally above and below the grid-point should be included, as these can (in principle at least)

indicate information leakage from higher altitudes that the elements directly above.

**P8L8:***"To classify this event, a comparison of all GW events in January 2016 has been performed in the operational analyses of ECMWF (Fig. 5)."*

How are the GW located in the ECMWF data and how are the occurrence frequencies calculated? a bit more details on how this was done would be good to include.

**P17L17:***"In Fig. A1 left column the instantaneous value $\xi_{z=11.5}$ at the middle point of the fitting volume of each individual ray is compared to an average value $\xi$ over the full height range of the S3D fitting volume (comparable to the S3D fitting result);"*

What is meant by full height range of the S3D fitting volume? Does it refer to each individual fitting volume i.e. 160 km x 160 km x 3.6 km?

––––––––––––––––––––––––––––––

---

## Referee Comment (RC2) · Anonymous Referee #2 · 15 Sep 2017

September 15, 2017

The manuscript describes a very interesting case study of atmospheric gravity waves using data from the limb imager onboard the aircraft. The tomographic method reveals the 3D propatgation of a gravity wave packet, the 3D ray-tracing simulation locates the potential wave sources and clearly reveals the horizontal propagation of GWs under the influence of background winds. The paper is overall well written and organized. It is suitable for publication after making the some revisions. Most of my comments are minor but some of them require more clarification and explanation.

1. Page 1, Line 2-3, the term 'global atmospheric models' see 6.

2. Page 1, Line 5, measured - observed/revealed.

3. Page 1, Line 12-13, the last sentence should be reworded

4. Page 1, Line 17, I donot see a good logic relation when using 'Thereby'.

5. Page 1, Line 20, add abbreviation QBO for quasi-biennial oscillation.

6. Page 2, Line 5, the term 'global atmospheric circulation models'. Even there are more than one options, mostly GCM refers to 'general circulation model'.

7. Page 2, Line 14, there are more published papers about the gravity wave parameterization schemes such as Alexander and Dunkerton 1999, Beres et al. 2004 and 2005, Richter et al. 2010.

8. Page 2, Line 15, besides the source distribution, the launched wave propagation direction is another simplification.

9. Page 2, Line 24, polarization - polarisation, analyzed - analysed, you may skip this since they are just differences between American and Bristish English.

10. Page 2, Line 25, there are several published papers using multiple instruments (colocated or network) to study the 3D strucuture of gravity waves such as Lu et al. 2016 (two lidar and imager), Cao et al. 2016 (lidar and imager), Bossert et al. 2015 (lidar and imager).

11. Page 3, Line 5, remove 'measurement',

12. Page 3, Line 7, the title of this section could better be 'Data and Methodolody'.

13. Page 3, Line 18, I suppose less pixels are used thus the readout time is reduced.

14. Page 3, Line 23, what is the aircraft flight altitude? what is the altitude range the measurements are taken?

15. Page 4, table 1 caption, the second sentence: The last column indicates the retrieved quantity for each spectral range.

16. Page 4, Line 10, structure - signatures.

17. Page 4, Line 12, this part - which part? the altitude range?

18. Page 5, Line 7, what is the temporal resolution, such as the integration time or exposure time?

19. Page 5, Line 9-10, what is the evidence that this is a mountain wave? This is important because this is the prerequisite of fitting. Is it stationary or near-stationary during your 2 hour observation window?

20. Page 5, Line 12, remove 'as discussed in Sec. 3.2'. It is not proper to refer to something in latter discussions.

21. Page 5, Line 17, 'GW'-'GWs'.

22. Page 5, Line 18-19, what is the relation between S-G filter and the polynomial fitting?

23. Page 6, Line 1, 'direction'-'directions', 'taken'-'treated'.

24. Page 6, Line 2, 'can be seen' - 'is demonstrated'.

25. Page 6, Figure 2, I feel Fig. 2 could be improved for better visulization. Add $x - y - z$ coordinate to show the scale of wave structures. The colorbar for positive and negative temperature perturbation should be properly chosen (red-white-blue) to clearly demonstrate the wave pattern. Figure 3 of Wright et al. 2017 is a good example.

26. Page 7, Figure 3, the x-coordinate of bottom sub-figures could be just the distance, which is more straightforward to compare the scales of GWs.

27. Page 7, Section 3 title 'Analysis'-'Results'.

28. Page 7, Line 8, and the data plotted in Fig. 2 and Fig. 3, what is the horizontal resolution of the raw temperature measurements? You may clarify these basic information in the text.

29. Page 7, Line 9, '3D direction'- please clarify this direction, is this direction the wave 'propagation direction' or the orientation of the wave front? You assume it is

a mounain wave, so the wave is not really 'propagating'. So if it is the orientation of the wave front, is there a relationship between the wave front orientation and the mountain ridge orientation?

30. Page 7, Line 14, 'the strength of the coupling of a GW with the background', what does this mean? Does it mean the same as the forcing/drag of GW.

31. Page 7, Line 15, at the end add 'when they dissipate.'

32. Page 8, Line 1, since those wave parameters are derived from fitting, are there confidence intervals that can describe the robustness of the fitting, say the uncertainties of those fitted parameters.

33. Page 8, Line 5-6, the total momentum in the order of GN ($10^9$N) seems to be a gigantic number, what is the physical meaning of this total momentum? The force the wave exerts on atmosphere? And how do you distinguish these two waves spatially?

34. Page 8, Line 7, how do you quantify the GW and calculate MF from ECWMF model?

35. Page 8, Line 8, how do you calculate this 0.14%? Do you mean the largest 0.14% of the all GWs?

36. Page 8, Line 14, 'characterize'-locate or identify.

37. Page 8, Line 15, 'advance'-'advantage'.

38. Page 8, Line 17, in this condition, when ray-tracing is discussed, GW intrinsic parameters rather than MF matter here.

39. Page 9, Figure 5, this is the intermittency of the gravity waves, which is mainly described by this probability distribution. I suppose you can make a similar plot

using the MF derived from your observations, which I think makes more sense to quantify the intermittency of the gravity waves retrieved from your observations. If go further, the log-normal distribution can also be fitted in the probability distributions.

40. Page 9, Line 9-10, for each dot of different size, it could be better visualization if you add a white edge for each dot, then they can be still visible when overlapped with dense trajectories.

41. Page 9, Line 10-11, 'according to the GWMF at the source location', so here you implicitly assume the GWs do not undergo any dissipation when they propagate from source to measurement locations?

42. Page 9, Line 13-14, what is the point of this 6 hour, in your Figure 6A, you indicate it is a 1-day backward simulation. So is there any conflict between these two? Then, can we understand this time is related to the propagating speed of the wave packet, say how much time it takes to propagate from source to measurement location. If so, a speed (group speed?) could be estimated.

43. Page 10, Line 1-2, the turning of the wave vectors could be explained by the wave refraction.

44. Page 11, the ray-tracing simulation (backward and forward) of GW propagation and the comparsion between 1D vs. 4D run are dramatically interesting and important. I expect more discussions about the ray-tracing results, especially on how this study can advance our understanding of the horizontal propagation of GWs and insights into GW parameterization.

45. Page 14, Line 16, $60°$.

46. Page 17, please skip the questions regarding the uncertainties of fitted GW parameters.

---

## Referee Comment (RC3) · Anonymous Referee #3 · 19 Sep 2017

This paper reports on first measurements obtained with a novel observation technique: limb imaging. This technique is applied here to obtain unprecedented observations of gravity waves in the lower stratosphere during one winter flight of the airborne GLORIA instrument above Iceland. The limb imaging technique and the associated tomographic retrieval allows the authors to provide for the first time an observational 3D view of a mountain wave packet over an extended mountain ridge, as well as to fully characterize the wave packet in terms of amplitude, horizontal and vertical wavelengths, as well as momentum flux.

The precise characterization of the gravity-wave packet enabled by GLORIA is further used to identify the wave source and to trace its forward propagation in the stratosphere, which convincingly underlines the significance of oblique wave propagation in

the atmosphere, a common defficiency of current gravity-wave drag parameterizations.

This article therefore provides a very clear demonstration of limb imaging capabilities for gravity-wave studies, and I support its publication with only minor revisions, which are detailed hereinunder.

**Minor comments**

- p4, l5: Could you be more specific here regarding the "regularization term" or provide a reference where the use of this term is better detailed?

- p4, l7: Similarly, could you be more specific on the smoothing you are using in the raw ECMWF fields?

- Section 2.2

  - Could you please state the airplane altitude during GLORIA measurements?

  - One primary goal of the article is to show how GLORIA observations can be used to accurately retrieve gravity-wave fluctuations. I am therefore surprised that you did not try to show comparisons between the retrieved 3D temperature field and in-situ observations performed by the airplane before the hexagonal path or with the dropsonde measurements, as well as with the resolved gravity-wave structures in the ECMWF analyses. In my opinion, such comparison should further support the capabilities of GLORIA, and perhaps also provide an additional way of characterizing the instrument performances.

- p7, l10: It may be worth stating that Equation (2) actually only applies in the so-called mid-frequency approximation, where "pseudo-momentum" and "momentum fluxes" are stricly equivalent. Otherwise, the sentence here may be slightly confusing.
I furthermore wonder whether this approximation is really valid in this case study. The ratio of horizontal/vertical wavelengths seems to imply relatively long waves, for which inertial effects in Equation (2) could not be totally neglected.

- Section 3.1, last paragraph: this comparison looks somewhat biased to me: if I have well understood, the GWMF for the Iceland case study are in one hand estimated from GLORIA observations, while in the other hand they are compared to a distribution of GWMF computed with ECMWF operational analyses. There is actually no garanty that ECMWF analyses accurately resolve such mountain wave events, and e.g. Jewtoukoff et al. (2015) have reported a significant underestimation of GWMF in ECMWF operational analyses.

**References**

Jewtoukoff, V., A. Hertzog, R. Plougonven, A. de la Camara, and F. Lott, Comparison of gravity waves in the southern hemisphere derived from balloon observations and the ECMWF analyses, *J. Atmos. Sci.*, **72**, 3449-3468, 2015.

---

## Author Comment (AC1) · 26 Oct 2017

Dear Mr Christensen,

Thank you very much for these very helpful comments!

According to comment #1 we will include details on the smoothing filter used for the generation of the a-priori field as well as on the calculation of the occurrence probability (comment #3). Comment #2 is a valuable hint and we will change the determination of the resolution. Instead of the sphere in the horizontal and the FWHM of the row of the averaging kernel in the vertical, we will now use a 3D ellipsoid to

estimate the resolutions in horizontal and vertical simultaneously. This new technique will account for the diagonal elements of the AVK matrix. However, applying the new method does not change the resulting resolution values. In the appendix we will clarify the sentence mentioned in comment #4.

A detailed list of all changes regarding these comments can be found below.

Again, thank you very much for helping us to present the theoretical background accurately and for improving the discussion and interpretation of results.

Sincerely, Isabell Krisch

**Reviewer comment:** What smoothing filter was used (type and fwhm) for the a-priori field generation?
**Authors response:** More details will be included in the text.
**Text changes:** This smoothing was done by applying a low-pass Fourier filter with cut-off wavenumber 18 in zonal direction. In height and latitude direction Savitzky-Golay (SG) filter (Savitzky and Golay, 1964) was applied with 4th order polynomials over 11 and 25 neighbouring points respectively. On the one hand, the so generated a-priori field improves the convergence speed of the iterative minimization, as this temperature structure is close to the true values due to the high quality of the ECMWF model. On the other hand, the smoothening ensures that any GW signature in the retrieval result does not stem from the used a priori data. If the a-priori data exerts any influence, it would dampen the GW structure.

**Reviewer comment:** Are you simply looking at the elements in the row of the AVK

matrix corresponding to the grid points directly above the grid-point of the row, or are you collapsing (i.e. performing a summation) the two other dimensions (x,y) before this FWHM is calculated? If the first approach is used, some description of the elements located diagonally above and below the grid-point should be included, as these can (in principle at least) indicate information leakage from higher altitudes that the elements directly above.

**Authors response:** The determination technique for the resolution will be changed to a 3D ellipsoidal fit to include diagonal matrix elements. However, applying the new method, does not change the resulting resolution values.

**Text changes:** The horizontal and vertical resolutions can be defined by the axes of the smallest ellipsoid containing all elements of the averaging kernel larger than half the maximum. Accordingly, in the middle of the performed hexagonal flight path, the vertical resolution is around 200 m, the horizontal resolution around 20 km.

**Reviewer comment:** How are the GW located in the ECMWF data and how are the occurrence frequencies calculated? a bit more details on how this was done would be good to include.

**Authors response:** We will include the details on how the occurrence probabilities are determined in the manuscript.

**Text changes:** To classify this event, a comparison of all GW events in January 2016 has been performed in the 6-hourly operational analyses of ECMWF. First the temperature background was isolated, as described in Sec. 2.1 for the a-priori field, and subtracted from the original field. The remaining temperature residuals were analyzed for GWs using the 3D sinusoidal fit algorithm described above. The GWMFs for all cubes were calculated. The GWMFs from all 124 analyses fields were combined to obtain the probability of GW occurrence (Fig. 6*, former Fig. 5*). Here, all GWMF values were considered independent of the horizontal and vertical wavelengths. Removing wavelengths larger than 2.5 times the cube size in order to filter less significant fits (not shown) induced no major changes in the general shape

of the distribution. This indicates that GW events with less certain fits do not bias the probability distribution. For the GW event over Iceland similar GWMF magnitudes were determined from the ECMWF analyses and from the GLORIA measurements. Thus, a comparison of the measurement results with the occurrence probability determined from the ECMWF analyses seems reasonable. According to Fig. 6 the measured GW event can be classified as a very strong case since the sum of all occurrence probabilities of stronger events is far below 1%.

**Reviewer comment:** What is meant by full height range of the S3D fitting volume? Does it refer to each individual fitting volume i.e. 160 km x 160 km x 3.6 km?
**Authors response:** The sentence will be restructured to clarify this point.
**Text changes:** In Fig. A1 left column the instantaneous value $\xi_{z=11.5}$ of the ray at the middle point of each fitting volume is compared to an average $\bar{\xi}$ of all values of the ray in the height range of the respective S3D fitting volume (comparable to the S3D fitting result).

**References**

Savitzky, A. and Golay, M. J. E.: Smoothing and Differentiation of Data by Simplified Least Squares Procedures., Analytical Chemistry, 36, 1627–1639, doi:10.1021/ac60214a047, http://dx.doi.org/10.1021/ac60214a047, 1964.

---

## Author Comment (AC2) · 26 Oct 2017

Dear Referee #2,

Thank you very much for these very helpful comments!

To increase the visualization of the paper, Figures 2 and 3 will be updated according to remarks #25 and #26. According to comment #33, the physical meaning of the total momentum, which is a measure for the drag, a GW event can exert on the atmosphere, will be better explained. This number is especially important as it can be compared to the GW drag of GCMs and cannot be provided by 1D wind observations. Comment

[Figure]

**34 has been mentioned in a similar way by all referees. We will include more details on how the occurrence probabilities were determined. A similar distribution derived from GLORIA measurements (comment #39), cannot be provided, as tomographic measurement patterns are event based only and, thus, not suitable for a statistical analysis. We very much appreciate remark #44 and will extend the discussion on the different results of 1D and 4D ray-tracing, respectively. Further we will include a paragraph on how the results of our paper can advance GW parameterizations.**

In addition, we will address all other minor comments in the paper. A detailed list of all changes can be found below.

Again, thank you very much for helping us to present the theoretical background accurately and for improving the discussion and interpretation of results.

Sincerely, Isabell Krisch

**Reviewer comment:** Page 1, Line 2-3, the term 'global atmospheric circulation models'. Even there are more than one options, mostly GCM refers to 'general circulation model'.
**Authors response:** All occurrences of 'global atmospheric circulation models' will be replaced by general circulation model.

**Reviewer comment:** Page 1, Line 5, measured - observed/revealed.
**Authors response:** This will be changed in the manuscript.

**Reviewer comment:** Page 1, Line 12-13, the last sentence should be reworded.
**Authors response:** The last sentence of the abstract will be reworded.
**Text changes:** Forward ray-tracing reveals that the waves propagate laterally more than 2000 km away from their source region. A comparison of a 3D ray-tracing version to solely column based propagation showed that lateral propagation can help the waves to avoid critical layers and propagate to higher altitudes. Thus, the implementation of oblique gravity wave propagation into general circulation models may improve their predictive skills.

**Reviewer comment:** Page 1, Line 17, I do not see a good logic relation when using 'Thereby'.
**Authors response:** The relation between the two sentences will be clarified.
**Text changes:** The exerted drag induces the meridional circulation in the mesosphere / lower thermosphere (MLT) and finally leads to the cold summer mesopause and the warm winter stratopause.

**Reviewer comment:** Page 1, Line 20, add abbreviation QBO for quasi-biennial oscillation.
**Authors response:** The abbreviation will be added.

**Reviewer comment:** Page 2, Line 14, there are more published papers about the gravity wave parameterization schemes such as Alexander and Dunkerton 1999, Beres et al. 2004 and 2005, Richter et al. 2010. Page 2, Line 15, besides the source distribution, the launched wave propagation direction is another simplification.
**Authors response:** A more detailed list of references on GW parameterization schemes will be included. The simplifications in the source distributions have been mentioned.

**Reviewer comment:** Page 2, Line 24, polarization - polarisation, analyzed - analysed, you may skip this since they are just differences between American and Bristish English.
**Authors response:** This will be addressed in the revised manuscript.

**Reviewer comment:** Page 2, Line 25, there are several published papers using multiple instruments (colocated or network) to study the 3D strucuture of gravity waves such as Lu et al. 2016 (two lidar and imager), Cao et al. 2016 (lidar and imager), Bossert et al. 2015 (lidar and imager).
**Authors response:** The mentioned further examples of 3D measurements in the mesosphere have been mentioned in the manuscript. However, these measurements have the same restriction as the MAARSY measurements, that they are far away from the sources and limited to few ground based stations.
**Text changes:** In the mesosphere, a full wave characterization of short scale GWs has been achieved with the Middle Atmosphere Alomar Radar System (MAARSY; Stober et al., 2013). For medium scale GWs in the mesosphere, a full characterization has been derived by combining lidar and airglow imager measurements (Bossert et al., 2015; Lu et al., 2015; Cao et al., 2016). However, all these observations are limited to a few ground based stations. Further, it is difficult to link observations at altitudes as high as the mesopause region to specific GW sources, which are usually located at much lower altitudes in the troposphere and lower stratosphere. So far no measurement technique existed to measure the 3D structure of mesoscale GWs in the lower stratosphere.

**Reviewer comment:** Page 3, Line 5, remove 'measurement'
**Authors response:** This word will be removed.

**Reviewer comment:** Page 3, Line 7, the title of this section could better be 'Data and

Methodolody'. Page 7, Section 3 title 'Analysis'-'Results'.
**Authors response:** Both section titles will be changed.

**Reviewer comment:** Page 3, Line 18, I suppose less pixels are used thus the readout time is reduced.
**Authors response:** This will be addressed in the text.

**Reviewer comment:** Page 3, Line 23, what is the aircraft flight altitude? what is the altitude range the measurements are taken?
**Authors response:** These points will be included in the manuscript.
**Text changes:** The aircraft flight altitude during this time was between 12.5 km and 13.5 km. Towards low altitudes, the GLORIA measurements were limited by clouds reaching up as far as 9 km to 10.5 km.

**Reviewer comment:** Page 4, table 1 caption, the second sentence: The last column indicates the retrieved quantity for each spectral range.
**Authors response:** This will be changed in the text.

**Reviewer comment:** Page 4, Line 12, this part - which part? the altitude range?
**Authors response:** This will be changed in the text.

**Reviewer comment:** Page 5, Line 7, what is the temporal resolution, such as the integration time or exposure time?
**Authors response:** A sentence on the temporal resolution will be added to Sec. 2.1.
**Text changes:** The time needed to accomplish the hexagon was about 2 h. During this time 2200 infrared images and corresponding spectra were taken. The presented tomographic retrieval represents a temporal mean over all these measurements.

**Reviewer comment:** Page 5, Line 9-10, what is the evidence that this is a mountain wave? This is important because this is the prerequisite of fitting. Is it stationary or near stationary during your 2 hour observation window?
**Authors response:** These waves were predicted by the ECMWF forecast to be stationary above Iceland for more than 6 hours. This will be included in the text.

**Reviewer comment:** Page 5, Line 12, remove 'as discussed in Sec. 3.2'. It is not proper to refer to something in latter discussions.
**Authors response:** This will be removed.

**Reviewer comment:** Page 5, Line 17, 'GW'-'GWs'.
**Authors response:** This will be changed.

**Reviewer comment:** Page 5, Line 18-19, what is the relation between S-G filter and the polynomial fitting?
**Authors response:** The method proposed by (**?**), which is called Savitzky-Golay filter, is based on a so-called running polynomial fitting. A polynomial of order k is fitted to a box of n neighbouring points and the middle point is replaced by the value of the fit. This is done for all data points by shifting the boxes over the data. Thus, Savitzky-Golay filter is a common name for polynomial filter.
**Text changes:** This was done through applying SG filters to the GLORIA temperature data in all three dimensions. For the SG filtering, 3rd order polynomials were fitted to 25, 60, and 60 neighbouring points in the vertical, zonal, and meridional directions.

**Reviewer comment:** Page 6, Figure 2, I feel Fig. 2 could be improved for better visulization. Add x - y - z coordinate to show the scale of wave structures. The

colorbar for positive and negative temperature perturbation should be properly chosen (redwhite-blue) to clearly demonstrate the wave pattern. Figure 3 of Wright et al. 2017 is a good example.
**Authors response:** Fig. 2 will be changed according to the remarks.

**Reviewer comment:** Page 6, Line 1, 'direction'-'directions', 'taken'-'treated'. Page 6, Line 2, 'can be seen' - 'is demonstrated'.
**Authors response:** This will be changed.

**Reviewer comment:** Page 7, Figure 3, the x-coordinate of bottom sub-figures could be just the distance, which is more straightforward to compare the scales of GWs.
**Authors response:** Fig. 3 D-F now plots distance as x-axis. Furthermore, the colormap of Fig. 3 will be changed to be in agreement with the updated Fig. 2.

**Reviewer comment:** Page 7, Line 8, and the data plotted in Fig. 2 and Fig. 3, what is the horizontal resolution of the raw temperature measurements? You may clarify these basic information in the text.
**Authors response:** This information can be found at the end of Sec. 2.1.

**Reviewer comment:** Page 7, Line 9, '3D direction'- please clarify this direction, is this direction the wave 'propagation direction' or the orientation of the wave front? You assume it is a mountain wave, so the wave is not really 'propagating'. So if it is the orientation of the wave front, is there a relationship between the wave front orientation and the mountain ridge orientation?
**Authors response:** The 3D direction is the direction of the wave vector $\vec{k} = (k, l, m)$. And the wave vector is oriented perpendicular to the mountain ridge. 2 panels with the horizontal wave vector direction and the orography of Iceland will be added to Fig. 5

(*former Fig. 4*).

**Text changes:** The fitted parameters are the 3D wave vector $\vec{k} = (k, l, m)$ and the amplitudes (Fig. 5C). Horizontal and vertical wavelengths, and the horizontal wave direction were calculated from the wave vector $\vec{k}$ and shown in Fig. 5A, 5B, and 5E, respectively. Fig. 5F shows the orography of Iceland. The main mountain ridge is oriented in east-west direction. As expected for a mountain wave, the horizontal wave direction (Fig. 5E) is perpendicular to the ridge orientation.

**Reviewer comment:** Page 7, Line 14, 'the strength of the coupling of a GW with the background', what does this mean? Does it mean the same as the forcing/drag of GW. Page 7, Line 15, at the end add 'when they dissipate.'

**Authors response:** The misleading expression 'strength of the coupling' will be replaced and the sentence will be reworded.

**Text changes:** Integrating the GWMF over the horizontal extent of a GW event leads to the total momentum, which determines the maximal drag this GW event can exert on the background flow in coupling and dissipation processes.

**Reviewer comment:** Page 8, Line 1, since those wave parameters are derived from fitting, are there confidence intervals that can describe the robustness of the fitting, say the uncertainties of those fitted parameters.

**Authors response:** A discussion of relevant effects for the uncertainty of the fitted parameters and the resulting confidence intervals is given in Appendix A. This will be mentioned here.

**Reviewer comment:** Page 8, Line 5-6, the total momentum in the order of GN (109N) seems to be a gigantic number, what is the physical meaning of this total momentum? The force the wave exerts on atmosphere? And how do you distinguish these two waves spatially?

**Authors response:** The total momentum corresponds to the maximal drag a GW event can exert on the background flow in coupling and dissipation processes. The two waves were defined to be north and south of 66.2 N. These points will be clarified in the text.

**Text changes:** The horizontal distribution of the GWMF clearly highlights two distinct wave packets: one with local GWMF of up to 50 mPa north of 66.2 N and one with local GWMF of up to 100 mPa south of 66.2 N. The GLORIA observations provide the horizontal variations of GWMF at 11.5 km altitude. This allows to integrate over the corresponding area of the two events and calculate the total momentum, a measure for the maximal drag this GW event can exert on the background flow in coupling and dissipation processes. This is a main advantage with respect to 1D wind observations, which can provide peak GWMF values but not the area for which these values are valid.

**Reviewer comment:** Page 8, Line 7, how do you quantify the GW and calculate MF from ECWMF model? Page 8, Line 8, how do you calculate this 0.14%? Do you mean the largest 0.14% of the all GWs?

**Authors response:** Similar points were mentioned by all three reviewers. We will include the details on how the occurrence probabilities are determined in the manuscript.

**Text changes:** To classify this event, a comparison of all GW events in January 2016 has been performed in the 6-hourly operational analyses of ECMWF. First the temperature background was isolated, as described in Sec. 2.1 for the a-priori field, and subtracted from the original field. The remaining temperature residuals were analyzed for GWs using the 3D sinusoidal fit algorithm described above. The GWMFs for all cubes were calculated. The GWMFs from all 124 analyses fields were combined to obtain the probability of GW occurrence (Fig. 6*, former Fig. 5*). Here, all GWMF values were considered independent of the horizontal and vertical wavelengths. Removing wavelengths larger than 2.5 times the cube size in order to filter less significant fits (not shown) induced no major changes in the general shape of the distribution. This indicates that GW events with less certain fits do not bias the probability distribution.

For the GW event over Iceland similar GWMF magnitudes were determined from the ECMWF analyses and from the GLORIA measurements. Thus, a comparison of the measurement results with the occurrence probability determined from the ECMWF analyses seems reasonable. According to Fig. 6 the measured GW event can be classified as a very strong case since the sum of all occurrence probabilities of stronger events is far below 1%.

**Reviewer comment:** Page 8, Line 14, 'characterize'-locate or identify. Page 8, Line 15, 'advance'-'advantage'.
**Authors response:** These points will be changed in the manuscript.

**Reviewer comment:** Page 8, Line 17, in this condition, when ray-tracing is discussed, GW intrinsic parameters rather than MF matter here.
**Authors response:** This will be changed in the text.

**Reviewer comment:** Page 9, Figure 5, this is the intermittency of the gravity waves, which is mainly described by this probability distribution. I suppose you can make a similar plot using the MF derived from your observations, which I think makes more sense to quantify the intermittency of the gravity waves retrieved from your observations. If go further, the log-normal distribution can also be fitted in the probability distributions.
**Authors response:** A similar distribution derived from GLORIA measurements cannot be provided, as tomographic measurement patterns are event based only and, thus, not suitable for a statistical analysis.

**Reviewer comment:** Page 9, Line 9-10, for each dot of different size, it could be better visualization if you add a white edge for each dot, then they can be still visible

when overlapped with dense trajectories.
**Authors response:** Fig. 7 will be updated in this respect.

**Reviewer comment:** Page 9, Line 10-11, 'according to the GWMF at the source location', so here you implicitly assume the GWs do not undergo any dissipation when they propagate from source to measurement locations?
**Authors response:** This will be addressed in the revised manuscript.
**Text changes:** These GWMF values are conservative estimates, as the backward ray-tracing cannot account for dissipation processes.

**Reviewer comment:** Page 9, Line 13-14, what is the point of this 6 hour, in your Figure 6A, you indicate it is a 1-day backward simulation. So is there any conflict between these two? Then, can we understand this time is related to the propagating speed of the wave packet, say how much time it takes to propagate from source to measurement location. If so, a speed (group speed?) could be estimated.
**Authors response:** We constructed a GROGRAT background atmosphere allowing for backward integration up to one day. However, the rays actually arrived at the mountains already after 6 hours. This will be clarified in the text.
**Text changes:** As can be seen in Fig. 7C, the ray-traces need between 3 and 6 hours to reach the ground. This is in good agreement with a vertical group velocity of 2 to 3 km/h, which has been calculated from the measurements. Hence, the GWs are probably excited roughly 6 hours before the measurements were taken.

**Reviewer comment:** Page 10, Line 1-2, the turning of the wave vectors could be explained by the wave refraction.
**Authors response:** This explanation will be added to the manuscript.
**Text changes:** Over Eastern Europe, the GWs are refracted by a horizontal wind shear, which changes their horizontal wave vector from southward to westward.

This allows the waves to quickly propagate upward into the westerly wind in the mid stratosphere.

**Reviewer comment:** Page 11, the ray-tracing simulation (backward and forward) of GW propagation and the comparsion between 1D vs. 4D run are dramatically interesting and important. I expect more discussions about the ray-tracing results, especially on how this study can advance our understanding of the horizontal propagation of GWs and insights into GW parameterization.

**Authors response:** The authors very much appreciate this comment and will extend the discussion, respectively.

**Text changes:** Two processes might play a significant role here: First, in the 1D GRO-GRAT version the GWs are not refracted and the wave vectors do not change its horizontal orientation with altitude. The westerly background winds at higher altitudes do not favor the propagation of GWs with wave vectors perpendicular to the wind direction. Second, in the full GROGRAT run, the GWs propagate horizontally away from the source. Hence, the GWs avoid the critical level positioned above the source location and more GWMF is transported to higher altitudes. Global mountain wave modeling (Xu et al., 2017) suggests that this effect may prevail also on a global basis.

Neither a realistic orientation of the wave vector, nor oblique GW propagation are incorporated in GW parameterizations used in current climate and weather prediction models (McLandress, 1998; Alexander and Dunkerton, 1999; Richter et al., 2010; McLandress et al., 2012; Garcia et al., 2017). However, both processes are context of several studies aiming to improve GW parameterizations (Preusse et al., 2009; Sato et al., 2009; Kalisch et al., 2014; Amemiya and Sato, 2016; Ribstein and Achatz, 2016; Garcia et al., 2017). The present paper provides a strong motivation to finally implement these processes in current climate and weather prediction models. Especially, as this could close gaps of GWMF in regions with sparse sources (McLandress et al., 2012) and reduce the cold-pole bias of climate and weather prediction models in the

lower stratosphere (Garcia et al., 2017).

**Reviewer comment:** Page 14, Line 16, 60.
**Authors response:** This will be changed in the text.

**Reviewer comment:** Page 17, please skip the questions regarding the uncertainties of fitted GW parameters.
**Authors response:** The authors think that the uncertainty discussion is an important part of this paper and therefore prefer to keep these paragraphs.

**References**

[revised manuscript text omitted]

3K
1.5K
0K
-1.5K
-3K

15km

12.5km

10km

2.5km

0km

67N
65N
63N
25W
15W
5W

**Fig. 1.**

[Figure]

[Figure]

**Fig. 2.**

[Figure]

**Fig. 3.**

---

## Author Comment (AC3) · 26 Oct 2017

Dear Referee #3,

Thank you very much for these very helpful comments!

According to comment #2 we will include details on the smoothing filter used for the generation of the a-priori field. Further, we will include a figure with a comparison of GLORIA and in-situ measurements (comment #4). As mentioned by the Referee, this will further support the capabilities of GLORIA. Regarding comment #5, the authors want to clarify, that for the derivation of Equation (2), the mid-frequency approximation

was not used. However, the low and high frequency terms of Equation (6) in Ern et al. (2004) have been omitted in our paper, as their impact on the result is below 1% in the measured gravity wave range. This will be addressed in the revised manuscript. Comment #6 has been mentioned in a similar way by all referees. We will include more details on how the occurrence probabilities are determined. A comparison of GLORIA measurements with ECMWF shows that for the case discussed in our paper, the GWMFs have a similar magnitude. Therefore, we have confidence that the statistics derived from ECMWF are a meaningful way to set our event into a broader context.

In addition, we will address all other minor comments in the paper. A detailed list of all changes can be found below.

Again, thank you very much for helping us to present the theoretical background accurately and for improving the discussion and interpretation of results.

Sincerely, Isabell Krisch

**Reviewer comment:** Could you be more specific here regarding the "regularization term" or provide a reference where the use of this term is better detailed?
**Authors response:** More details on the used regularization can be found in Ungermann et al. (2010). This citation will be mentioned in the text.

**Reviewer comment:** Similarly, could you be more specific on the smoothing you are using in the raw ECMWF fields?
**Authors response:** More details will be included in the text.
**Text changes:** This smoothing was done by applying a low-pass Fourier filter with cutoff wavenumber 18 in zonal direction. In height and latitude direction Savitzky-Golay (SG) filter (Savitzky and Golay, 1964) was applied with 4th order polynomials over 11 and 25 neighbouring points respectively. On the one hand, the so generated a-priori field improves the convergence speed of the iterative minimization, as this temperature structure is close to the true values due to the high quality of the ECMWF model. On the other hand, the smoothening ensures that any GW signature in the retrieval result does not stem from the used a priori data. If the a-priori data exerts any influence, it would dampen the GW structure.

**Reviewer comment:** Could you please state the airplane altitude during GLORIA measurements?
**Authors response:** This point will be included in the manuscript.
**Text changes:** The aircraft flight altitude during this time was between 12.5 km and 13.5 km. Towards low altitudes, the GLORIA measurements were limited by clouds reaching up as far as 9 to 10.5 km.

**Reviewer comment:** One primary goal of the article is to show how GLORIA observations can be used to accurately retrieve gravity-wave fluctuations. I am therefore surprised that you did not try to show comparisons between the retrieved 3D temperature field and in-situ observations performed by the airplane before the hexagonal path or with the dropsonde measurements, as well as with the resolved gravity-wave structures in the ECMWF analyses. In my opinion, such comparison should further support the capabilities of GLORIA, and perhaps also provide an additional way of characterizing the instrument performances.
**Authors response:** A comparison of the GLORIA measurements with in-situ measurements and ECMWF will be included as a new figure (Fig. 4). Both measurements agree well within the spatial resolution range of GLORIA.
**Text changes:** Fig. 4 shows a comparison of the retrieval results with in-situ measurements and ECMWF operational analyses with T1279/L137 resolution. The retrieval results and model data were interpolated onto the in-situ measurement locations. The GLORIA measurements agree well with the in-situ measurements. Some very short scales are beyond the spatial resolution of GLORIA. The ECMWF analysis catches the main variations, but the temperature oscillations are not as strong as in reality. GLORIA can better reproduce peaks as for example the one around 10:40 UTC. This comparison underlines the high quality of the GLORIA measurement data.

**Reviewer comment:** It may be worth stating that Equation (2) actually only applies in the socalled mid-frequency approximation, where "pseudo-momentum" and "momentum fluxes" are stricly equivalent. Otherwise, the sentence here may be slightly confusing. I furthermore wonder whether this approximation is really valid in this case study. The ratio of horizontal/vertical wavelengths seems to imply relatively long waves, for which inertial effects in Equation (2) could not be totally neglected.

**Authors response:** For the derivation of Equation (2), the mid-frequency approximation was not used. However, the low and high frequency terms of Equation (6) in Ern et al. (2004) have been omitted as their impact on the result is below 1% in the measured gravity wave range. A full discussion of the importance of these low and high frequency terms can be found in Ern et al. (2017). This point will be clarified in the manuscript.

**Text changes:** Low and high frequency terms are omitted here due to simplicity. Deviations from the full equations derived by Ern et al. (2004) are less than 1% in the observational range of GLORIA. For a full discussion of the relevance of all correction terms see the supporting information in Ern et al. (2017).

**Reviewer comment:** Section 3.1, last paragraph: this comparison looks somewhat biased to me: if I have well understood, the GWMF for the Iceland case study are in one hand estimated from GLORIA observations, while in the other hand they are compared to a distribution of GWMF computed with ECMWF operational analyses. There is actually no garanty that ECMWF analyses accurately resolve such mountain wave events, and e.g. Jewtoukoff et al. (2015) have reported a significant underestimation of GWMF in ECMWF operational analyses.

**Authors response:** This point was mentioned by all three reviewers. We will include the details on how the occurrence probabilities are determined in the manuscript. A comparison of GLORIA measurements with ECMWF shows that for the case discussed in our paper, the GWMFs have a similar magnitude. Therefore, we have confidence that the statistics derived from ECMWF are a meaningful way to set our event into a broader context.

**Text changes:** To classify this event, a comparison of all GW events in January 2016 has been performed in the 6-hourly operational analyses of ECMWF. First the temperature background was isolated, as described in Sec. 2.1 for the a-priori field, and subtracted from the original field. The remaining temperature residuals were analyzed for GWs using the 3D sinusoidal fit algorithm described above. The GWMFs for all cubes were calculated. The GWMFs from all 124 analyses fields were combined to obtain the probability of GW occurrence (Fig. 6*, former Fig. 5*). Here, all GWMF values were considered independent of the horizontal and vertical wavelengths. Removing wavelengths larger than 2.5 times the cube size in order to filter less significant fits (not shown) induced no major changes in the general shape of the distribution. This indicates that GW events with less certain fits do not bias the probability distribution.

For the GW event over Iceland similar GWMF magnitudes were determined from the ECMWF analyses and from the GLORIA measurements. Thus, a comparison of the measurement results with the occurrence probability determined from the ECMWF analyses seems reasonable. According to Fig. 6 the measured GW event can be classified as a very strong case since the sum of all occurrence probabilities of stronger events is far below 1%.

**References**

Ern, M., Hoffmann, L., and Preusse, P.: Directional gravity wave momentum fluxes in the stratosphere derived from high-resolution AIRS temperature data, Geophys. Res. Lett., 44, 475–485, doi:{10.1002/2016GL072007}, 2017.

Ern, M., Preusse, P., Alexander, M. J., and Warner, C. D.: Absolute values of gravity wave momentum flux derived from satellite data, Journal of Geophysical Research: Atmospheres, 109, doi:10.1029/2004JD004752, http://dx.doi.org/10.1029/2004JD004752, 2004.

Jewtoukoff, V., Hertzog, A., Plougonven, R., de la Camara, A., and Lott, F.: Comparison of gravity waves in the southern hemisphere derived from balloon observations and the ecmwf analyses. Journal of the Atmospheric Sciences, 72(9):3449–3468, doi:10.1175/JAS-D-14-0324.1, https://doi.org/10.1175/JAS-D-14-0324.1, 2015.

Savitzky, A. and Golay, M. J. E.: Smoothing and Differentiation of Data by Simplified Least Squares Procedures., Analytical Chemistry, 36, 1627–1639, doi:10.1021/ac60214a047, http://dx.doi.org/10.1021/ac60214a047, 1964.

Ungermann, J., Kaufmann, M., Hoffmann, L., Preusse, P., Oelhaf, H., Friedl-Vallon, F., and Riese, M.: Towards a 3-D tomographic retrieval for the air-borne limb-imager GLORIA, Atmos. Meas. Tech., 3, 1647–1665, doi:10.5194/amt-3-1647-2010, 2010.

**[ACPD](ACPD)**
[Figure]

[Figure]

**Fig. 1.**

[Figure]